# Development of Autonomous Robot Osteotomy for Mandibular Ramal Bone Harvest and Evaluation of Its Accuracy: A Phantom Mandible-Based Trial

Ik Jae Kwon [1], Soung Min Kim [1,*] and Soon Jung Hwang [2,*]

1 Department of Oral and Maxillofacial Surgery, School of Dentistry and Dental Research Institute, Seoul National University, Seoul 03080, Korea; ijkwon@snu.ac.kr

2 Dental Research Institute, Seoul National University, HSJ Dental Clinic for Oral and Maxillofacial Surgery, Seoul 06626, Korea

* Correspondence: smin5@snu.ac.kr (S.M.K.); sjhwang@snu.ac.kr (S.J.H.);
Tel.: +82-2-2072-0213 (S.M.K.); +82-2-595-4737 (S.J.H.); Fax: +82-2-766-4948 (S.M.K.); +82-2-525-4738 (S.J.H.)

**Featured Application: Unlike long-bone surgery, in jaw bone, which is a relatively small surgical field, the operation is significantly disturbed with a bulky device. In this study, an autonomous robot arm osteotomy with direct coordinate determination for registering was suggested and may provide a basis for developing clinically application.**

**Abstract:** An autonomous robot osteotomy using direct coordinate determination for registering was developed, and the accuracy of the designed osteotomy along the preprogrammed plan was evaluated. Furthermore, the accuracy of the robotic and manual osteotomy was compared in regard to cut position, length, angle and depth. A light-weight robot was used in this study, with an electric gripper. Twenty stone models were used to evaluate accuracy of osteotomy and sixteen mandible phantoms were used to simulate the ramal bone harvest osteotomy for comparison between robotic and manual surgery. In the stone model experiment, the absolute mean values for osteotomy errors for position, length, angle, and depth were $0.93 \pm 0.45$ mm, $0.81 \pm 0.34$ mm, $1.26 \pm 1.35°$, and $1.19 \pm 0.73$ mm, respectively. In the mandible phantom model experiment, the robotic surgery showed lower errors for position, length and angle ($0.70 \pm 0.34$ mm, $0.35 \pm 0.19$ mm and $1.32 \pm 0.96°$) and somewhat higher errors for depth ($0.59 \pm 0.46$ mm) than manual surgery ($1.83 \pm 0.65$ mm, $0.62 \pm 0.37$ mm, $5.96 \pm 3.47°$ and $0.40 \pm 0.31$ mm). This study may provide a basis for developing clinical application of an autonomous robot osteotomy.

**Keywords:** autonomous robot; three-points coordinate determination; robot surgery; accuracy; osteotomy; ramal bone harvest



## 1. Introduction

Medical robotics has tremendous potential for improving accuracy and precision when performing surgical procedures. In recent decades, medical robots have helped doctors in the operating room by doing tasks difficult to perform with human eyes and hands, and are developing rapidly. Depending on the degree of user interaction, three categories of robot systems are defined, direct or manual control, shared control and supervisory control robotic systems [1]. In direct control, the surgeon operates the slave robot directly through the master console. In shared control, the surgeon and controller share the manipulator command and work together in order to carry out a task. In supervisory control, the procedure is executed solely by the robot, which acts according to a computer program that the surgeon inputs prior to the procedure. In the other classification, two system groups can be distinguished in the field of surgical robotics [2]. The first group, telemanipulators, is not preprogrammed and moves exactly as controlled by a slave console. Within the other group, preprogrammed surgical robots execute a preoperatively defined trajectory [2].

First-generation surgical robots consisted mainly of robotic arms designed to assist the primary surgeon by holding and positioning instruments such as a laparoscopic camera or retractor. Surgical robots have transcended the role of assistant to become the primary surgeon's hands through a computer interface [3]. The representative model, da Vinci Surgical System (Intuitive Surgical, Inc., Sunnyvale, CA, USA), incorporates three-dimensional (3D) stereoscopic vision with two or three robotic slave arms equipped with instruments that have six degrees of freedom and wrist-like motions.

The direct control robots, which are represented by da Vinci, were controlled manually while viewing the screen directly by the operator, so the robots do not need to automatically determine the position of the patient. However, in the autonomous robot, the robot determines the patient coordinates, and the operation is performed based on these coordinates. Therefore, how to effectively register the patient's coordinates is an important issue for autonomous robots. In this study, we used a robot arm guided registration technique, which is one of a direct coordinate determination system by three points.

Because osteotomy is the most commonly used technique for various operations in oral and maxillofacial surgery [4], the osteotomy design was selected for our study design. Among the many osteotomies, osteotomy for ramal bone graft (RBG) is relatively difficult to approach with good visibility. Moreover, osteotomy for RBG includes cutting in various directions using various instruments [5]. Long-bone osteotomy, such as for a fibula free flap or Le Fort I osteotomy have better accessibility for the robot than the osteotomy for RBG, and are simple one-direction osteotomy examples [6,7]. Therefore, we needed to evaluate RBG osteotomy accuracy in various directions and categories.

Autogenous bone harvesting from the mandibular ramus is the first choice for reconstruction of maxillofacial defects. The mandibular ramus area has many advantages over other donor sites in the oral cavity [5]. Although the incidence of donor site complications is rare in RBG, clinicians have made efforts to reduce potential side effects. Side effects include sensory disturbances or mandibular fractures, and can lead to unnecessary patient suffering as well as significant stress for the operating surgeons. The use of robots in an osteotomy of a RBG could lead to a more accurate osteotomy and less complications, resulting in greater patient and surgeon satisfaction. However, currently, there are no commercially available robots in the field of oral and maxillofacial surgery, and there are not many ongoing studies. Autonomous robots have not yet been developed to cut the jaw bone with burs or saws.

Various instruments such as rotary motor, laser, piezoelectric device [8] and waterjet [9] can be used during jaw osteotomy. Among these, the basic tools commonly used include a rotary motor with bur or saw. There are various bur types including round burs, diamond burs and fissure burs. Also, there are various types of saws, such as reciprocating and oscillating. In this study, osteotomy was performed with a fissure bur and disc saw because of the convenience for cutting bone.

It is not easy to compare an osteotomy performed by a robot and one performed manually. Most previous studies compared robotic and manual surgeries using only qualitative methods or with indirect outcomes such as operative time [8,10]. An autonomous robotic osteotomy system for RBG was developed in our study, and the accuracy of the designed osteotomy along a preprogrammed plan was evaluated. In addition, we compared the position, length, angle and depth errors of the cuts between robot arm and human.

## 2. Materials and Methods

### 2.1. Overview of Robotic System

A KUKA light-weight robot (LBR iiwa 7 R800, KUKA Robotics, Augsberg, Germany) was used to position the gripper and dental handpiece. This robot arm with seven degrees of freedom was connected to an electric two-finger parallel gripper (SCHUNK GmbH & Co. KG, Lauffen am Neckar, Germany). The customized flange of the robot arm was designed for connecting the gripper and robot. The dental handpiece with the fissure bur and 20 mm

diameter disc saw was held by the gripper tightly. The surgeon changed the fissure bur and disc during the operation manually (Figure 1).

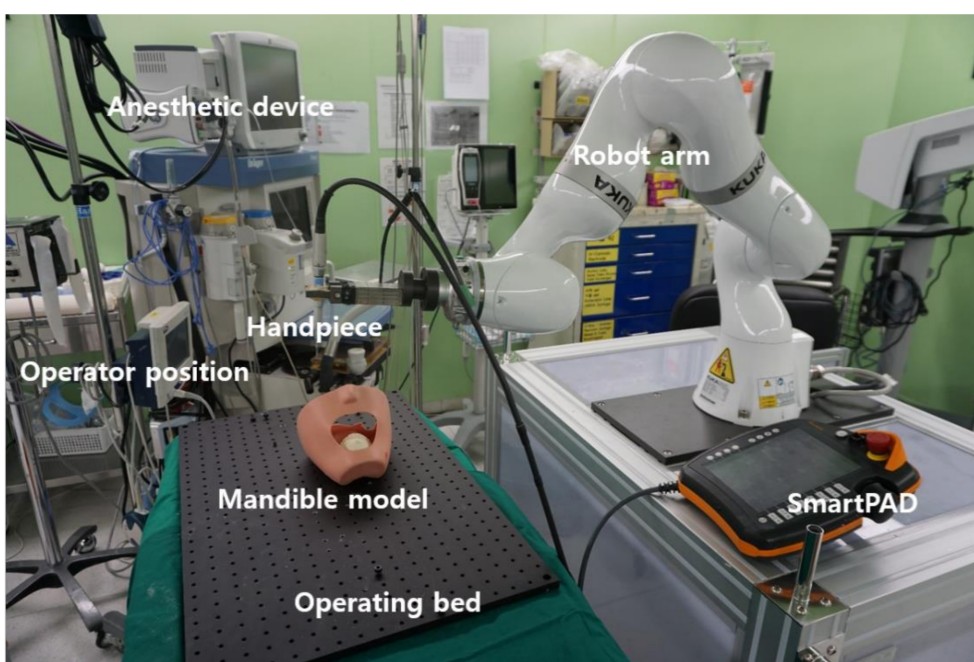

**Figure 1.** Overview of our autonomous robot osteotomy system.

The robot consisted of four parts: (a) Workbench notebook, (b) Robot controller (cabinet), (c) Robot arm, (d) SmartPAD®.

An application was programmed using the workbench notebook. After uploading the program to the robot controller, we ran the application on the robot arm. It was possible to operate the robot through the smartPAD®, and the robot could be operated manually by the smartPAD® or automatically by the preprogrammed application.

A gripper was connected to the end of the robot arm, and this gripper held a dental low-speed handpiece connecting a disc saw or fissure bur for bone cutting. We used two different tools with a surgical fissure bur (2 mm diameter and 15 mm length cutting edge) and two kinds of discs (6 mm and 20 mm diameter with 0.5 mm thickness).

*2.2. Three-Points Direct Coordinate Determination*

In most robot systems involving motion, registration of preoperative 3D imaging and the actual patient or model is required. Our robot arm could detect its position and joint angle accurately within 0.1 mm. In this study, the robotic arm held the instrument and directly touched three reproducible points on the teeth of the mandible model to achieve registration.

The robot uses a Cartesian coordinate system and the radians of $\alpha$, $\beta$ and $\gamma$ are the value of Euler angles rotated by the z-, y- and x-axes, respectively. Euler angles are three angles introduced by Leonhard Euler to describe the orientation of a rigid body with respect to a fixed coordinate system [11]. Any orientation can be achieved by composing three elemental rotations.

When we refer to the order of the unit coordinate transformations, the complex coordinate transformation matrix from the m-coordinate system to the i-coordinate system can be seen as follows.

$$C_m^i = \begin{pmatrix} cos\beta cos\alpha & sin\gamma sin\beta cos\alpha - cos\gamma sin\alpha & cos\gamma sin\beta cos\alpha + sin\gamma sin\alpha \\ cos\beta sin\alpha & sin\gamma sin\beta sin\alpha + cos\gamma cos\alpha & cos\gamma sin\beta sin\alpha - sin\gamma cos\alpha \\ -sin\beta & sin\gamma cos\beta & cos\gamma cos\beta \end{pmatrix} \qquad (1)$$

Likewise, Equation (1) shows the complex coordinate transformation matrix from the m-coordinate system to the i-coordinate system.

One Cartesian coordinate system can be determined by three different points in space that are not on the same line. We used a robot arm holding the instrument to detect three points in space and use these 3D positioning values to calculate the new Cartesian coordinate system to match the preoperative image and actual position.

When each of the three reference points were detected, the robot calculated the model coordination step by step. The first point represents the origin of the new coordinate. The second point is used for determining the x-axis. The direction from the first point to second point is the direction of the x-axis. Finally, the third point is used for determining the xy-plane by including all three points. The z-axis is defined as an axis passing through the origin and perpendicular to the xy-plane and calculated by cross product of the x-axis and y' vector, which is from first point to third point. The y-axis is defined automatically by the z- and y-axes and calculated by the cross product of them (Figure 2).

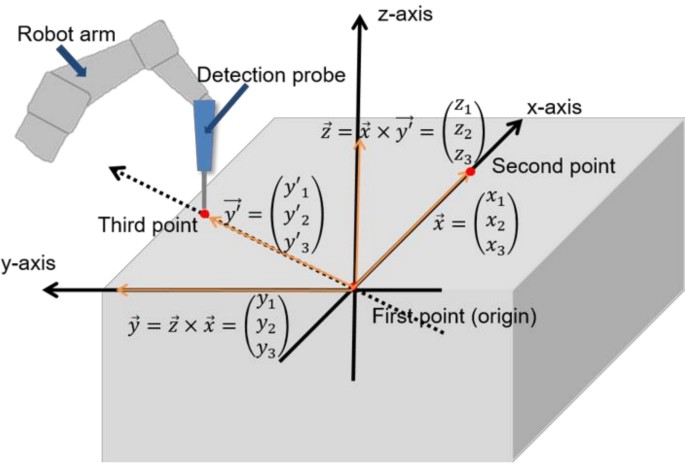

**Figure 2.** Overview of the three-points coordinate determination using Euler angle conversion.

### 2.3. Autonomous Robot Osteotomy on Stone Model

### 2.3.1. Study Design

Twenty rectangular stone models (7 cm × 7 cm × 3 cm) were designed for the test bench. To use direct coordinate determination by three points, we touched the three different points on the upper plane of the model by hand, manipulating the robot arm gripping the pointer. The robot was set on impedance mode, which modulates the joint impedance. By decreasing the joint impedance, the robot could be manipulated easily by hand. After coordinate determination, the robot arm automatically determined the osteotomy position. The osteotomy line was designed similar to the RBG line (2 cm × 1 cm × 0.5 cm), and divided into posterior, anterior, inferior, and superior cuts. The posterior and anterior line was cut by a fissure bur. The posterior cut was performed first with a 10 mm position and 5 mm length, and the anterior cut was performed similarly, 20 mm apart from the posterior line. Subsequently, the tool was manually changed to the disc saw to cut the inferior and superior lines. At the inferior cut, the disc saw was located anteriorly first and achieved a 5 mm depth, and then proceeded to the posterior direction with 20 mm length along a straight line. At the last superior cut, the robot arm was rotated three-dimensionally for proper positioning and cutting the 20 mm line with a 5 mm depth (Figure 3a).

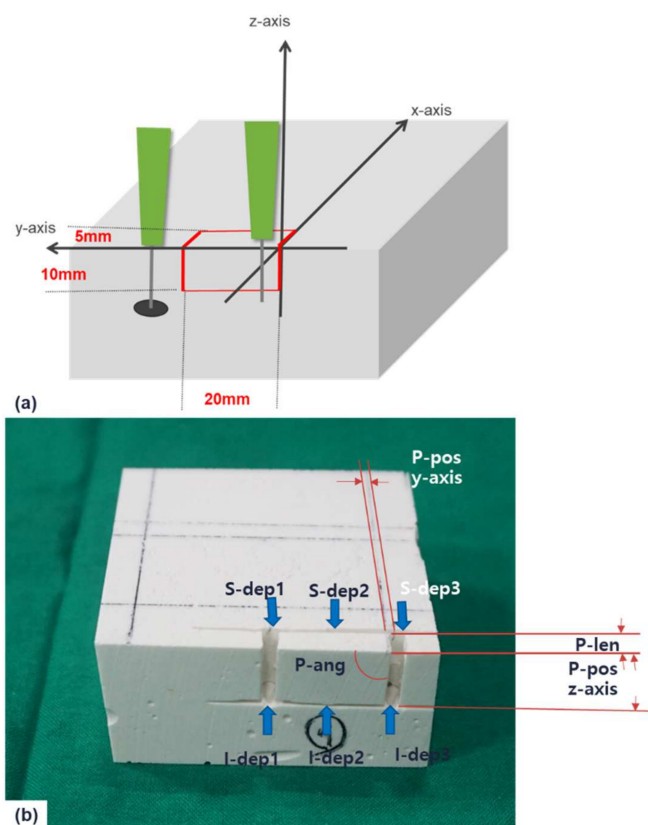

**Figure 3.** Design of model ramal bone graft osteotomy (**a**), measurements of model ramal bone graft osteotomy. P-pos: posterior position error, P-len: posterior length error, P-ang: posterior angle error, I-dep: inferior depth error, S-dep: superior depth error (**b**).

### 2.3.2. Accuracy Evaluation

Each position, length, angle, and depth of the four cuts (posterior, anterior, inferior, and superior) were measured by an electronic caliper and goniometer. Errors of position, length, angle, and depth for accuracy evaluation of the osteotomy cutting line were considered by calculating the difference between the planned values and results. The position error was defined as the difference between the preplanned position and the actual robot arm position. The position error on the x-axis could not be measured, because the robot approached the test model and utilized its position as a reference point when it came into contact with the surface during the osteotomy procedure. So the position error could be measured on only the y- and z-axes. The posterior cut started at the origin of the coordinate system. Length error was defined as the differences in sliding fissure bur cuts of the anterior and posterior parts from the preplanned 5 mm. Angle error was defined as the angle differences between the bur and disc within 90 degrees. Finally, depth error was defined as indicating the depth difference of inferior and superior disc cuts. The length differences were measured at two points, top and bottom. In the inferior and superior cuts, position and angle were measured at the two points, anterior and posterior, and depth was measured at the three points: anterior, middle and posterior. Absolute mean values for all error types were calculated and the root mean square (RMS) of the position error was obtained additionally (Figure 3b).

### 2.4. Comparison of Robot and Manual RBG Osteotomy on Mandible

#### 2.4.1. Design for 3D Virtual Planning

A total of sixteen mandible phantoms (Pacific Research Laboratories Inc., Vashon, WA, USA) were used to simulate RBG osteotomy. A 3D image of a phantom was obtained by 3D scanning and a virtual RBG was performed using the MIMICS 19.0 software (Materialise,

Leuven, Belgium). To perform the RBG osteotomy, the robot should know the transformation from the mandible coordinate to the ramus coordinate. A coordinate of the phantom model was determined by three-point coordinate determination. The three points could be detected easily (Figure 4a): (a) The distal edge of the mandibular left lateral incisor, (b) The buccal pit of the left second molar, (c) The lingual fossa of the right second molar.

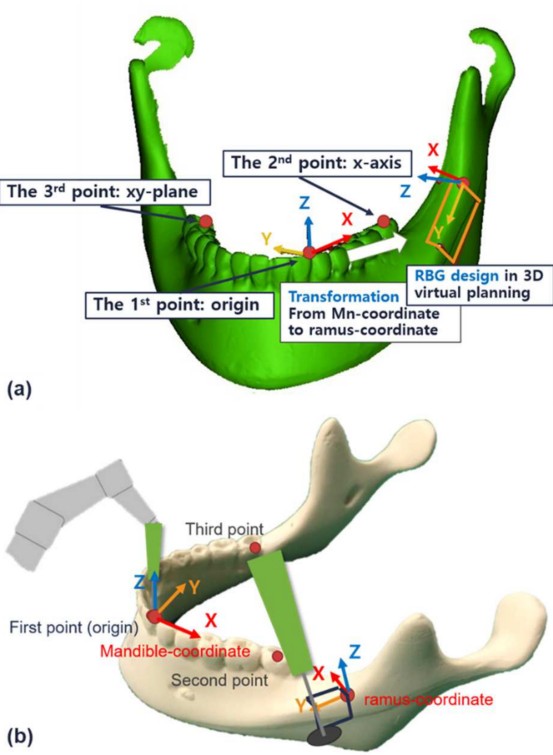

(a)

(b)

**Figure 4.** Schematic for the three-point coordinate determination and coordinate transform for the mandible phantom model. The ramal bone graft osteotomy was designed on the left ramus area (**a**). Schematics of the three-point coordinate determination and ramal bone graft osteotomy on the mandible phantom (**b**).

The RBG was designed with a virtual model with a size of 2 cm × 1 cm and a thickness of only 0.3 cm (2 cm × 1 cm × 0.3 cm). For RBG mandible osteotomy, one more transformation was needed for transformation from the phantom coordinate to the ramus coordinate. A two-transformation matrix was calculated from Equation (1) and multiplied to obtain the final transformation matrix. After solving the equation, the final position of the robot arm (X, Y, Z, α, β and γ) based on the ramus coordinate could be obtained. Finally, the robot could detect the ramus coordinate and perform the RBG osteotomy on the preprogrammed position (Figure 4b).

On the left side, the robot surgery was performed as described above, and on the right side, a conventional manual surgery was performed in the same order as the robotic surgery. For the manual surgery, surgical instruments and measuring tools from a typical actual surgery were used in an operating room. A surgical ruler and pencil were used to design the RBG osteotomy. The osteotomy started at the fixed reference point, which was defined as the point 29 mm from the lingual groove of the right second molar and the anterior ramus. After that, the other osteotomy line was traced so that it mirrored that on the left side (2 cm × 1 cm × 0.3 cm).

For the posterior and anterior cuts, the surgery was done with a fissure bur in hand. For a 10 mm cut, the fissure bur moved into the model 3 mm deep on both sides. For the inferior and superior cuts, we divided the robot surgery group into two groups: 6 mm diameter disc and 20 mm diameter disc. Among the sixteen total mandible phantoms, eight were in the 6 mm disc group and eight were in the 20 mm disc group. For all sixteen

phantoms, the left side was operated on by the robot and the right side was operated on manually.

### 2.4.2. Comparison and Evaluation

After RBG osteotomy on both ramus, cone beam computer tomography (CBCT; Point 3D Combi 500, Pointnix Inc., Seoul, Korea) was used to evaluate the osteotomy. The CBCT images were reconstructed to form 3D images using MIMICS software. On the software, a preplanned design was aligned to the CBCT image so we could measure the differences in position, length, angle and depth of the cuts made for the RBG osteotomies. Each position, length, angle and depth of the four kinds of cuts (posterior, anterior, inferior and superior) were measured using the 3D software.

The position error on the x-axis could not be measured because the robot approached the test model and took its position as a reference point when it came into contact with the surface during the osteotomy. Thus, the position error could only be measured on the y- and z-axes. For measuring the depth of the inferior and superior cuts, we sliced the mandible model at the position of the posterior and anterior osteotomy on the software. At these sections, we could easily measure the depth of the cut. For the inferior and superior cuts, the position and angle were measured at two points: anterior and posterior, and the depth was also measured at two points: anterior and posterior. The absolute mean values for all the errors were calculated and the RMS of the position error was additionally obtained. These parameters were measured on both sides using the same method (Figure 5).

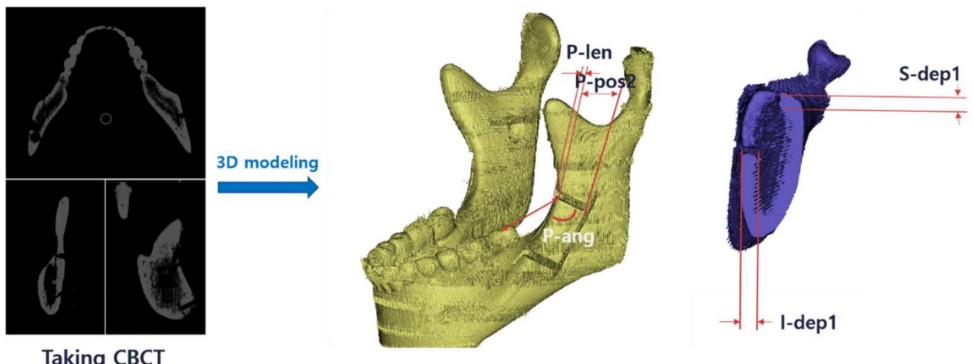

**Figure 5.** Accuracy evaluation of the mandible ramal bone graft osteotomy using CBCT and 3D modeling with MIMICS software. P-pos: posterior position error, P-len: posterior length error, P-ang: posterior angle error, I-dep: inferior depth error, CBCT: cone beam computer tomography.

Statistical analysis was performed using the statistical package SPSS for Windows (release 21.0.0.0, SPSS Inc., Chicago, IL, USA). Descriptive statistics for mean error and RMS of position, angle and length measurements are presented as the mean $\pm$ standard deviation (SD). The results of the robotic surgery group and manual surgery group were compared by *t*-test. A *p*-value of 0.05 or less was regarded as statistically significant.

## 3. Results

### 3.1. Flow Diagram

Our autonomous robotic osteotomy system for a ramal bone graft was divided into three main actions: referencing, stone RBG osteotomy and mandible RBG osteotomy. In the referencing machine, the position values of the three reference points detected by the robot arm were gathered and the data were sent to the three-points coordinate determination machine. In the direct coordinate determination machine, x, y, z, $\alpha$, $\beta$ and $\gamma$ for a transformation from robot coordinate to model coordinate were calculated using Euler angle transformation. The robot could then recognize the coordinate of the model and perform the model osteotomy as per the preprogrammed plan. At the third machine, the mandible RBG osteotomy, one more transformation module from the mandible coordinate to the

ramus coordinate was added. Thus, before doing the mandible ramal bone graft osteotomy, two Euler transformations are obtained. By combining these two transformations, the robot could determine final coordinates for the ramal bone harvest osteotomy (Figure 6).

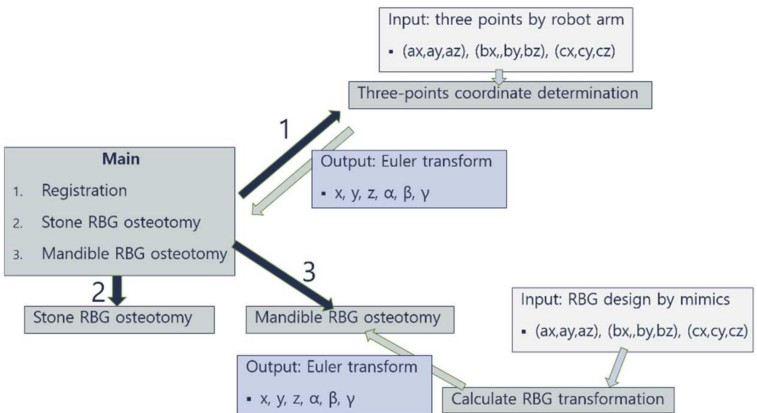

**Figure 6.** Workflow diagram for the autonomous robot osteotomy system.

### 3.2. Autonomous Robot Osteotomy on Stone Model and Its Accuracy

The registration of each of the twenty rectangular models was done by only three reference points using the robotic arm. After the direct coordinate determination, the robot could perform the RBG operation by itself successfully. The robot could automatically determine the position of origin and start the RBG osteotomy at that position. The robot moved along the x-axis and stopped when it touched the model surface. All instruments, including the fissure bur and 20 mm disc, could be programmed identically such that when the surface was detected, the handpiece was turned on to cut the model. At first, a fissure bur was used for posterior and anterior cuts, and then the instrument was changed to the disc saw to cut inferiorly and superiorly.

A total of twenty models were used to test the osteotomy using a robot arm. Each error for the RBG osteotomy model is summarized in Table 1 by cut type. The absolute mean values for osteotomy errors for position, length, angle, and depth were $0.93 \pm 0.45$ mm, $0.81 \pm 0.34$ mm, $1.26 \pm 1.35°$, and $1.19 \pm 0.73$ mm, respectively. The position error was estimated by the direction of the y- and z-axes, and RMS value was calculated. There was a significant difference between each group. The position error was significantly lower than the angle and depth errors. Similarly, the length error was significantly lower than the angle and depth errors (Table 2).

### 3.3. Comparison between the Robotic and Manual RBG Osteotomy on Mandible

Shown as a flow diagram in Figure 6, the transformation module was needed for mandible RBG osteotomy. The direct coordinates were determined using three points on the teeth, and these were used to register the mandible phantom model to the robot. The three reference points on the mandibular teeth included the distal edge of the mandibular left lateral incisor, the buccal pit of the left second molar and the lingual fossa of the right second molar. To reduce the registration error, these three reference points were defined as the points that could be taken with the greatest distance from each other in the oral cavity. The three points were detected in order, and then the mandible coordinate was determined. Next, the robot calculated the transformation automatically and positioned itself at the origin of the ramus coordinate. The RBG osteotomy could start at these points using the same algorithm as was used in a previous model RBG osteotomy.

**Table 1.** The absolute mean value of errors for position, angle, length and depth on model RBG osteotomy.

| Model RBG Osteotomy (*n* = 20) | | | Error (Mean ± SD) |
|---|---|---|---|
| Posterior cut | Position (mm) | RMS | 0.85 ± 0.28 |
| | Angle (Degree) | | 1.10 ± 1.24 |
| | Length (mm) | Top | 0.71 ± 0.27 |
| | | Bottom | 0.93 ± 0.35 |
| Anterior cut | Position (mm) | RMS | 1.16 ± 0.55 |
| | Angle (Degree) | | 1.54 ± 1.60 |
| | Length (mm) | Top | 0.74 ± 0.39 |
| | | Bottom | 0.86 ± 0.33 |
| Inferior disc cut | Position (mm) | RMS | 1.14 ± 0.48 |
| | Angle (Degree) | Anterior | 1.00 ± 0.96 |
| | | Posterior | 0.67 ± 0.74 |
| | Depth (mm) | Anterior | 0.75 ± 0.30 |
| | | middle | 0.69 ± 0.42 |
| | | Posterior | 0.76 ± 0.30 |
| Superior disc cut | Position (mm) | RMS | 0.68 ± 0.36 |
| | Angle (Degree) | Anterior | 1.95 ± 1.61 |
| | | Posterior | 1.32 ± 1.49 |
| | Depth (mm) | Anterior | 1.81 ± 0.68 |
| | | middle | 1.71 ± 0.66 |
| | | Posterior | 1.45 ± 0.82 |

RBG: ramal bone graft; SD: standard deviation, RMS: root mean square.

**Table 2.** Accuracy evaluation due to the category of errors on model RBG osteotomy.

| Category | | Error (Mean ± SD) |
|---|---|---|
| Position (mm) | RMS | 0.93 ± 0.45 |
| Length (mm) | | 0.81 ± 0.34 |
| Angle (Degree) | | 1.26 ± 1.35 |
| Depth (mm) | | 1.19 ± 0.73 |
| *p* value | | 0.000 |
| Post hoc (*p* < 0.05) | | Position-Angle<br>Position-Depth<br>Length-Angle<br>Length-Depth |

SD: standard deviation.

We performed a total of sixteen mandibular phantom surgeries using robotic arm on the left side and by manual on the right side with the same phantom model. Each category of errors for the mandible RBG osteotomy by the type of cuts is summarized in Table 3. The position error was measured by y- and z-axes and calculated as RMS values. Comparing the 6 mm disc and 20 mm disc groups, depth error was significantly higher on the 20 mm disc group than on the 6 mm disc group ($p = 0.009$), only on the superior cut of the robotic surgery (Table 3). Comparing robotic surgery and manual surgery, there were significant differences in absolute mean value and variance in all categories (Table 4). For robotic surgery, the position, length, angle and depth errors were 0.70 ± 0.34 mm, 0.35 ± 0.19 mm, 1.32 ± 0.96° and 0.59 ± 0.46 mm, respectively. For the manual surgery, the position, length, angle and depth errors were 1.83 ± 0.65 mm, 0.62 ± 0.37 mm, 5.96 ± 3.47° and 0.40 ± 0.31 mm, respectively. The robotic surgery had significantly higher accuracy and lower variance for position, length and angle errors. On the other hand, the absolute mean value and variance of the depth error was significantly higher for the robot surgery (Table 4).

**Table 3.** The absolute mean values of osteotomy errors for position, angle, length and depth on mandible phantoms categorized by 6 mm disc and 20 mm disc in the robotic surgery group and the 6 mm disc and 20 mm disc in the manual surgery group.

| Cut | Category | | Errors of Robot Surgery 6 mm Disc | 20 mm Disc | *p* Value | Errors of Hand Surgery 6 mm Disc | 20 mm Disc | *p* Value |
|---|---|---|---|---|---|---|---|---|
| Post. cut | Position (mm) | RMS variance | 0.65 ± 0.17 0.03 | 0.59 ± 0.22 0.05 | 0.602 | 1.93 ± 0.37 0.13 | 1.60 ± 0.51 0.26 | 0.347 |
| | Angle (°) | Mean ± SD Variance | 0.72 ± 0.50 0.25 | 1.81 ± 1.29 1.67 | 0.754 | 6.13 ± 3.75 14.07 | 8.10 ± 2.67 7.11 | 0.057 |
| | Length (mm) | Mean ± SD Variance | 0.40 ± 0.14 0.02 | 0.33 ± 0.20 0.04 | 0.599 | 0.71 ± 0.26 0.07 | 0.91 ± 0.44 0.19 | 0.602 |
| Ant. cut | Position (mm) | RMS Variance | 1.00 ± 0.24 0.06 | 1.28 ± 0.22 0.05 | 0.251 | 2.03 ± 0.54 0.29 | 2.53 ± 0.42 0.17 | 0.347 |
| | Angle (°) | Mean ± SD Variance | 0.93 ± 0.62 0.38 | 1.17 ± 0.69 0.47 | 0.117 | 7.15 ± 3.15 9.90 | 1.74 ± 1.46 2.14 | 0.602 |
| | Length (mm) | Mean ± SD Variance | 0.40 ± 0.24 0.06 | 0.29 ± 0.13 0.02 | 0.347 | 0.39 ± 0.27 0.07 | 0.48 ± 0.24 0.06 | 0.463 |
| Inf. cut | Position (mm) | RMS Variance | 0.36 ± 0.32 0.10 | 0.52 ± 0.18 0.03 | 0.602 | 2.29 ± 0.51 0.26 | 2.65 ± 0.62 0.38 | 0.754 |
| | Angle (°) | Mean ± SD Variance | 0.29 ± 0.65 0.42 | 1.58 ± 0.96 0.92 | 0.753 | 7.58 ± 2.99 8.94 | 7.23 ± 3.55 12.63 | 0.754 |
| | Depth (mm) | Mean ± SD Variance | 0.36 ± 0.35 0.12 | 0.45 ± 0.30 0.09 | 0.530 | 0.32 ± 0.24 0.06 | 0.51 ± 0.33 0.11 | 0.251 |
| Sup. cut | Position (mm) | RMS Variance | 0.86 ± 0.45 0.20 | 0.53 ± 0.31 0.09 | 0.602 | 0.79 ± 0.20 0.04 | 1.09 ± 0.41 0.16 | 0.465 |
| | Angle (°) | Mean ± SD Variance | 1.45 ± 1.08 1.17 | 1.30 ± 1.10 1.20 | 0.917 | 4.09 ± 2.80 7.83 | 5.29 ± 2.52 6.35 | 0.602 |
| | Depth (mm) | Mean ± SD Variance | 0.44 ± 0.21 0.05 | 1.13 ± 0.44 0.19 | 0.009 | 0.30 ± 0.25 0.06 | 0.46 ± 0.36 0.13 | 0.173 |

RBG: ramal bone graft; SD: standard deviation; Post. cut: Posterior cut; Ant. cut: Anterior cut; Inf. cut: inferior cut; Sup. cut: Superior cut.

**Table 4.** Comparison of accuracy between robotic and manual surgeries for position, length, angle and depth errors.

| | | Robot Surgery | Hand Surgery | *p* Value |
|---|---|---|---|---|
| Position | RMS variance | 0.70 ± 0.34 0.11 | 1.83 ± 0.65 0.43 | <0.001 <0.001 |
| Length | Mean ± SD variance | 0.35 ± 0.19 0.03 | 0.62 ± 0.37 0.14 | 0.009 0.004 |
| Angle | Mean ± SD variance | 1.32 ± 0.96 0.93 | 5.96 ± 3.47 12.08 | <0.001 <0.001 |
| Depth | Mean ± SD variance | 0.59 ± 0.46 0.21 | 0.40 ± 0.31 0.10 | 0.030 0.014 |

*3.4. Ergonomics and Safety*

We tested the ease with which the posterior mandibular ramal area could be approached by the robot arm osteotomy through an intraoral approach in the dummy operating theater. The surgeons' natural position could be guaranteed and enough space was provided (Figure 7). The vertical cut with the long fissure bur and the inferior cut were successfully simulated in the direct direction toward the operating site. For the superior cut, the robot arm should be rotated to horizontal. We set the midpoint of robot position so that, when the robot turned, it did not hit the patient or the surgeon.

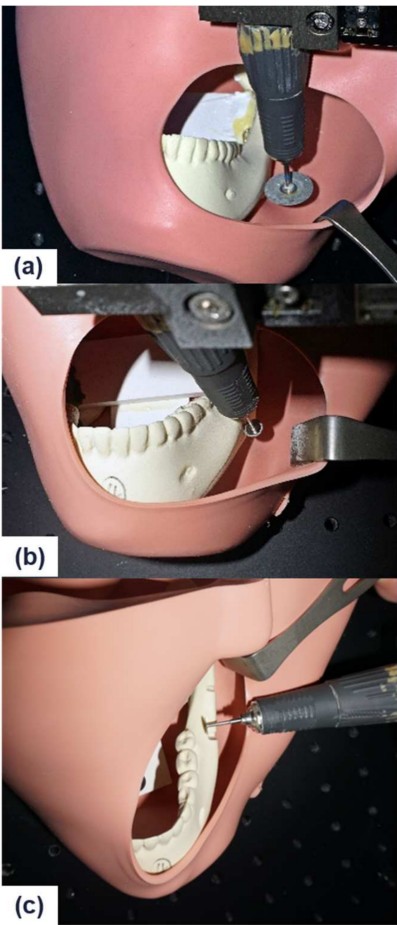

**Figure 7.** Test for the ease with which the posterior mandibular ramal area could be approached by the robot arm osteotomy through an intraoral approach in the dummy operating theater: inferior cut with 20 mm disc (**a**) and 6 mm disc (**b**), and superior cut with 6 mm disc (**c**).

The ergonomic aspects and safety features of robot guidance were assessed and confirmed under optimal conditions. The computer-assisted, robot arm osteotomy was compact enough to allow two surgeons to operate comfortably. To ensure safety of the patient, the robot would automatically stop if the robot arm was subjected to a torque of 15 N or more. Also, the handpiece engine was only manually operated. Our robot system is equipped with an emergency button on the smartPAD® that allows the robot to stop when an unexpected event occurs.

## 4. Discussion

An autonomous robot osteotomy scheme was developed and demonstrated with direct coordinate determination by three points on the teeth. The accuracy of our autonomous robotic osteotomy system was evaluated on both stone models and mandible phantoms. Some researchers reported that the RMS differences for an optical navigation between the planning and postoperative computed tomographic model were $1.31 \pm 0.28$ mm and $1.74 \pm 0.73$ mm, respectively [12]. In another study using skull fiducial or laser registration, the results had a $1.75 \pm 0.94$ mm radial error, $2.82 \pm 1.1$ mm depth error, and $3.39 \pm 1.078$ mm target point error [13]. In our system using direct coordinate determination, the position, length, angle, and depth errors were $0.93 \pm 0.45$ mm, $0.81 \pm 0.34$ mm, $1.26 \pm 1.35°$, and $1.19 \pm 0.73$ mm, respectively. Although the angle and depth errors were larger than the position and length errors, our system was accurate compared to autonomous robot studies using other registration methods.

At the posterior and anterior cuts, robot osteotomy was designed such that the fissure bur was sliding in the model by 5 mm as the depth of RBG design. The length error represented the accuracy of this action. Moreover, at the inferior and superior cuts, the 20 mm diameter disc saw moved into the model with a 5 mm depth and a slide with 20 mm length. Depth error represented the accuracy of how deep the disc moved into the model. These depth and angle errors were quite large compared with the position and length errors. The position error is position-dependent when the robot arm moves without any resistance, but the depth errors are affected by the changing resistance depending on the density of the model, which corresponds to bone in an actual patient. Therefore, the depth error tended to be larger than the other errors. The control of angular deviation is a different issue from positioning the end point. Our results demonstrated that the angular control of the end effector could be more difficult than position control.

In 1991, Taylor et al. performed the first orthopedic surgery for hip replacement using the ROBODOC surgical device, which was the first system that could implement a preplanned milling trajectory [14]. In oral and maxillofacial surgery, the first application of robot-assisted surgery was by Kavanagh, who performed preclinical tests of antrostomy using the ROBODOC system [15]. In 1998, the OTTO system (Surgical Robotics Laboratory, Medical Faculty Charité, Humboldt-University, Berlin, Germany) was developed as the first interactive robotic system for positioning an electric drill in maxillofacial surgery [16]. To the best of our knowledge, no robotic system has been specifically designed for cranio-maxillofacial reconstruction, particularly in hard tissue surgery [6]. In particular, there are no commercially available robots in the oral and maxillofacial surgery field, and there are not many ongoing studies.

Six levels of autonomy for medical robotics were proposed as one possible framework. (Level 0: No autonomy, Level 1: Robot assistance, Level 2: Task autonomy, Level 3: Conditional autonomy, Level 4: High autonomy, and Level 5: Full autonomy) [17]. Robot-assisted surgery and robot-guided surgery belongs to Level 1, Robot assistance. Our robot arm belongs to Level 2, Task autonomy, as it can partially perform tasks automatically. This study is a new attempt to use an autonomous robot in the maxillofacial region and goes one step further to a higher level of autonomy.

Mandible RBG osteotomy on a phantom model was performed successfully. There was no significant difference between the 6 mm disc group and the 20 mm disc group during both robotic and manual surgery. This means that there was no difference in the accuracy of the kinds of tools when either the robot or the person was operating. We recognized that the instrument was detecting the surface of the mandible model and made it work accordingly, so both 6 mm and 20 mm discs should work with the same program, and the accuracy with the different kinds of instruments should not be significantly different. However, comparing the robotic and manual surgeries, there were significant differences in many categories.

The error was calculated by dividing by the absolute mean value and the RMS. The accuracy of a measurement system is the degree of closeness of a quantity to true value. The precision of a measurement system, related to reproducibility and repeatability, is the degree to which repeated measurements under unchanged conditions show the same results. Considering these metrics, robotic surgery was better than manual surgery for the position and length errors in both accuracy and precision. The position error is an indicator of how accurately the end effector of the robot is located and the length error is an indicator of how accurately the end effector of the robot has moved during osteotomy of the model. Thus, obviously the robot can be positioned and moved more accurately and precisely than a person's hand. In the angle error, there was a different result. The absolute mean value of angle error was not significantly different between robotic and manual surgery, but the RMS of the angle error was significantly different between the groups. There was no difference in accuracy between the two groups in adjusting the angles to position the instruments. In this study, only the angle of 90° was used for the osteotomy design. The 90° angle can be measured relatively accurately by the human

naked eye. Therefore, there seems to be no difference in accuracy between the two groups. If we designed it at an arbitrary angle, such as 30°, 45° or 60° rather than 90°, the results might have been different. In regard to precision, the robot was better than humans for positioning at the designated angle. However, the depth error has the opposite results. The manual surgery had more accurate and precise results than the robotic surgery for the depth error, which was the largest of all the errors in our results. The depth control of the disc was the most difficult osteotomy type performed in this study. In the future, we need an algorithm that can feed back the resistance of the robot while it performs osteotomy in order to achieve more accurate depth control.

Safety is one of the most important issues, and our robot has several integrated safety features. One of them is protection of the patient and the surgeon against a patient's unexpected movement. When the robot is touched by external force more than 15 N in the surgical field, it automatically stops moving. As an amount of external force, we can modify the threshold of unpredicted external force. The ergonomic aspects of the robot guidance were assessed and confirmed under optimal conditions. The robot arm osteotomy was compact enough to allow two surgeons to operate comfortably. In the future, the robot will be able to play the role of one surgeon in the operating room.

Since a force of up to 15 N can be applied to the fissure bur (2 mm width and 15 mm length) connected to the robot arm, a pressure of up to 500 kPa can be applied to the bone. When this mechanical impact is applied to the bone, it can affect the interaction between the cell and the extracellular matrix [18]. Since this can affect bone healing, it will be important to control the robot arm carefully so that minimal pressure can be applied to reduce this bias.

Future research should be focused on the improvement of the real-time interaction between the end-effect of the robot and the target patient. More investigations are also needed to measure the dynamic data of ablated bone tissue, and for real-time monitoring and control of the depth of the disc or saw cut. This real-time interaction will add another important safety feature. In this study, it was found that direct coordinate recognition by three points on the teeth using the robotic arm can be useful in autonomous robot surgery. Atypical anatomic condition such as multiple supernumerary teeth [19] or cysts [20] may have impact on the design and conduction of osteotomies. They can be excluded in the design, or the size and shape of osteotomies should be changed. In addition, further research should focus on how to track patient movement after registration. We can track a designated point with a 3D camera or laser scanner and achieve the same effect as a navigation system with simpler equipment. Also, by using another robot arm connected to the patient's head for tracking, we can determine the movement of the patient's head position directly. This tracking method remains an additional challenge, and engineers and surgeons need to work together to overcome it.

## 5. Conclusions

An autonomous robot osteotomy scheme was developed, using the direct coordinate determination by three points on the teeth. The robotic surgery showed high accuracy and precision in positioning and somewhat low accuracy in controlling the depth of the disc sawing. Comparing robotic and manual surgeries, the robotic surgery was superior in accuracy and precision in position, length and angle control. However, the manual surgery had higher accuracy and precision in depth control. Therefore, if robot osteotomy will be applied to the actual clinical practice, it will be possible to make less invasive surgery and reduce postoperative swelling and pain.

**Author Contributions:** Conceptualization, S.J.H.; methodology, I.J.K.; validation, S.M.K.; formal analysis, I.J.K.; investigation, S.M.K.; resources, S.J.H.; data curation, I.J.K.; writing—original draft preparation, I.J.K.; writing—review and editing, S.J.H.; visualization, I.J.K.; supervision, S.M.K. and S.J.H. All authors have read and agreed to the published version of the manuscript.

**Funding:** This work was supported by a grant no. 01-2017-0010 from the SNUDH Research Fund.

**Institutional Review Board Statement:** Not applicable.

**Informed Consent Statement:** Not applicable.

**Data Availability Statement:** Not applicable.

**Conflicts of Interest:** The authors declare no conflict of interest.

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
