# Peer review of "Development of Autonomous Robot Osteotomy for Mandibular Ramal Bone Harvest and Evaluation of Its Accuracy: A Phantom Mandible-Based Trial"

_applsci, doi:10.3390/app11062885_

Round 1

Reviewer 1 Report

This is an interesting first approach for the development of an autonomous robot for mandibular bone harvest. The proof of concept seems validated on this in vitro model as the robot is more accurate for most parameters (except for drilling depth) compared to manual surgery. However the absolute differences are limited and the worst results show a difference of 1 to 2mm of deviation compared to planning.

My first remark goes to the precision requested for such osteotomy in the mandibular ramus aiming to harvest a bone block for autologous grafting: it seems that the precision requested is not that important as the bone block will be secondary shaped manually? The cost/benefice of such device is questionable as the harvesting of bone blocks in the ramus is done in routine by trained oral surgeons, with limited surgical complications. In the introduction, the necessity for developing such device should be more justified.

Page 2: Regarding the methods to harvest Ramus Bone for Autologous grafts, you have not mentioned Piezzosurgery ??

This is an in vitro study and the mandible is attached on the working platform: how do you plan to stabilize the patients’mandible to operate safely? Do you anticipate that it will be necessary to work under General Anesthesia for this kind of surgeries?

Nothing is said regarding the mucoperiosteal flap to access the ramus bone: this is an important point as it is critical to carefully design, raise and maintain the flap out of the surgical field while the bur is working: could you show a picture showing how the surgeon would manage the surgery together with the robot ?

The details of the different calculations could probably be simplified and shortened.

Author Response

Reviewer #1: Comments to the Author

My first remark goes to the precision requested for such osteotomy in the mandibular ramus aiming to harvest a bone block for autologous grafting: it seems that the precision requested is not that important as the bone block will be secondary shaped manually? The cost/benefice of such device is questionable as the harvesting of bone blocks in the ramus is done in routine by trained oral surgeons, with limited surgical complications. In the introduction, the necessity for developing such device should be more justified.

Answer> Thanks for your kind comments. We agree that osteotomy for ramal bone harvesting may not require much precision. The purpose of this study was not for the development of an alternative method against the conventional manual ramal bone harvest, but this study is the first step for autonomous robot osteotomy in oral and maxillofacial field. The way for the development of autonomous robot osteotomy, which can be applied in patients, are long and far from the first step, there are many barriers to be overcome, such as fixation of patient’s head or simultaneous 3D positional change of robot arm, prompt stop action for safety. The reason we chose ramal bone harvesting was that it contains osteotomy in various directions so that various factors such as position, length angle and depth could be evaluated.

Page 2: Regarding the methods to harvest Ramus Bone for Autologous grafts, you have not mentioned Piezzosurgery ??

Answer> Yes, a piezoelectric device is one of popularly used convenient devices for ramal bone harvesting. We considered it as one of the devices we could test in future studies, but in this study we used the most basic instrument, a rotary motor.

This is an in vitro study and the mandible is attached on the working platform: how do you plan to stabilize the patients’mandible to operate safely? Do you anticipate that it will be necessary to work under General Anesthesia for this kind of surgeries?

Answer> The purpose of this study was not for the development of an alternative method against the conventional manual ramal bone harvest, but this study is the first step for autonomous robot osteotomy in oral and maxillofacial field. The way for the development of autonomous robot osteotomy, which can be applied in patients, are long and far from the first step, there are many barriers to be overcome, such as fixation of patient’s head or simultaneous 3D positional change of robot arm, prompt stop action for safety., The experiment was performed under the condition that the mandible was fixed through the bite block and the skull was fixed on the table. If a tracking system such as navigation, a system for simultaneous 3D positional change of robot arm according to the head or mandibular movements and a safety system with prompt stop actioncan be used in combination with this system in the future, it can be used without fixing the mandible and skull.

Nothing is said regarding the mucoperiosteal flap to access the ramus bone: this is an important point as it is critical to carefully design, raise and maintain the flap out of the surgical field while the bur is working: could you show a picture showing how the surgeon would manage the surgery together with the robot?

Answer> Thanks for your valuable comments. We understand that realistic simulation and safety are very important. This study is the first step for autonomous robot osteotomy in oral and maxillofacial field. The way for the development of autonomous robot osteotomy, which can be applied in patients, are long and far from the first step, there are many barriers to be overcome. In this experiment, we evaluated only osteotomy procedure. We assumed that the operator elevated the mucoperiosteal flap manually and retracted the soft tissue. The added Figure 7 and explanation show how soft tissue was simulated and retracted during the actual experiment. After retraction of the soft tissue, the movement of robot arm was designed to enter the mouth without damaging the soft tissue.

The details of the different calculations could probably be simplified and shortened.

Answer> Yes, in the tables, the length was reduced by displaying only the RMS value.

Reviewer 2 Report

This is a very interesting and well conducted study on the potentials of the clinical application of robotic surgery for osteotomy.

I do not have any remarks or suggestions for improvement of the manuscript.

Author Response

Thanks for your kind comments.

Reviewer 3 Report

The article “Development of autonomous robot osteotomy for mandibular ramal bone harvest and evaluation of its accuracy: A Phantom Mandible-Based Trial” has merits and falls within the journal's scope.

The study has interesting novel information related to a novel autonomous  robot  for the osteotomies. Furthermore, the accuracy of the robotic and manual osteotomy was compared in regard to cut position, length, angle and dep.

This study basically has the merit to apply both concepts of biorobotics and biology.

The introduction should also improve some key-concepts:

  1. The authors must discuss on mechanical impact of biomaterials used on tissues and cells (Please, see “Tatullo M, Marrelli M, Falisi G, Rastelli C, Palmieri F, Gargari M, Zavan B, Paduano F, Benagiano V. Mechanical influence of tissue culture plates and extracellular matrix on mesenchymal stem cell behavior: A topical review. Int J Immunopathol Pharmacol. 2016 Mar;29(1):3-8.”) - (Please, see and also discuss “1. Marrelli M, Codispoti B, Shelton RM, Scheven BA, Cooper PR, Tatullo M, Paduano F. Dental Pulp Stem Cell Mechanoresponsiveness: Effects of Mechanical Stimuli on Dental Pulp Stem Cell Behavior. Front Physiol. 2018 Nov 26;9:1685.”)
  2. The authors must describe if atypical anatomic conditions may have impact on their technique (Cite and discuss as an example the supernumerary teeth “Inchingolo F, Tatullo M, Abenavoli FM, Marrelli M, Inchingolo AD, Gentile M, Inchingolo AM, Dipalma G. Non-syndromic multiple supernumerary teeth in a family unit with a normal karyotype: case report. Int J Med Sci. 2010 Nov 5;7(6):378-84.”)
  3. The authors must finally translate the main key-concepts of this novel concept of therapy and translational medicine in potential clinical applications, reporting if some pharmacological approach may impact on their study outcomes (Please, see and discuss on “Inchingolo F, Tatullo M, Marrelli M, Inchingolo AM, Picciariello V, Inchingolo AD, Dipalma G, Vermesan D, Cagiano R. Clinical trial with bromelain in third molar exodontia. Eur Rev Med Pharmacol Sci. 2010 Sep;14(9):771-4. PMID: 21061836.”)

Author Response

Reviewer #3: 

  1. The authors must discuss on mechanical impact of biomaterials used on tissues and cells (Please, see “Tatullo M, Marrelli M, Falisi G, Rastelli C, Palmieri F, Gargari M, Zavan B, Paduano F, Benagiano V. Mechanical influence of tissue culture plates and extracellular matrix on mesenchymal stem cell behavior: A topical review. Int J Immunopathol Pharmacol. 2016 Mar;29(1):3-8.”) - (Please, see and also discuss “1. Marrelli M, Codispoti B, Shelton RM, Scheven BA, Cooper PR, Tatullo M, Paduano F. Dental Pulp Stem Cell Mechanoresponsiveness: Effects of Mechanical Stimuli on Dental Pulp Stem Cell Behavior. Front Physiol. 2018 Nov 26;9:1685.”)`

Answer> Thanks for your kind comments. We agree that mechanical impact used on bone and cells is important factors to be considered. We added one paragraph as below in the Discussion section and an additional reference which was recommended.

“Since a force of up to 15 N can be applied to the fissure bur (2 mm width and 15 mm length) connected to the robot arm, a pressure of up to 500kPa can be applied to the bone. When this mechanical impact is applied to the bone, it can affect the interaction between the cell and the extracellular matrix [19]. Since this can affect bone healing, it will be important to control the robot arm carefully so that minimal pressure can be applied to reduce this bias.“

  1. The authors must describe if atypical anatomic conditions may have impact on their technique (Cite and discuss as an example the supernumerary teeth “Inchingolo F, Tatullo M, Abenavoli FM, Marrelli M, Inchingolo AD , Gentile M, Inchingolo AM, Dipalma G. Non-syndromic multiple supernumerary teeth in a family unit with a normal karyotype: case report. Int J Med Sci. 2010 Nov 5;7(6):378-84.”)

Answer> Thank you for the comment. A new sentence was added.

“Atypical anatomic condition such as multiple supernumerary teeth [20] or cysts [21] may have impact on the design and conduction of osteotomies. They can be excluded in the design, or the size and shape of osteotomies should be changed.”

  1. The authors must finally translate the main key-concepts of this novel concept of therapy and translational medicine in potential clinical applications, reporting if some pharmacological approach may impact on their study outcomes (Please, see and discuss on “Inchingolo F, Tatullo M, Marrelli M, Inchingolo AM, Picciariello V, Inchingolo AD, Dipalma G, Vermesan D, Cagiano R. Clinical trial with bromelain in third molar exodontia. Eur Rev Med Pharmacol Sci. 2010 Sep;14(9):771-4. PMID: 21061836.”)

Answer> Robot osteotomy is more accurate than freehand, so osteotomy can be performed less invasively without unnecessary bone removal, thus reducing many complications. Therefore, if this will be applied to the actual clinical practice, it will be possible to reduce postoperative swelling and pain. However, an additional pharmacological approach is not needed. We added a sentence below in the Conclusion section.

“Therefore, if robot osteotomy will be applied to the actual clinical practice, it will be possible to make less invasive surgery and reduce postoperative swelling and pain.”

Round 2

Reviewer 1 Report

Thank you for the instructive answers to my previous comments

Reviewer 3 Report

to accept